# Effects of Moso Bamboo (*Phyllostachys edulis*) Forest Stand Density on Root Growth and Soil Quality for Shoot Production Under a Long-Term Bamboo-Stocking Retention Model

**DOI:** 10.3390/biology14091179

**Published:** 2025-09-02

**Authors:** Tianyou He, Xing Cai, Jialin Zhang, Zongming Cai, Qingzhuan Chen, Shikun Li, Jing Ye, Lingyan Chen, Jundong Rong, Liguang Chen, Yushan Zheng

**Affiliations:** 1College of Landscape Architecture and Art, Fujian Agriculture and Forestry University, Fuzhou 350002, China; hetianyou@fafu.edu.cn (T.H.); jingye@fafu.edu.cn (J.Y.); fafucly@fafu.edu.cn (L.C.); 2College of Forestry, Fujian Agriculture and Forestry University, Fuzhou 350002, China; 22404028003@fafu.edu.cn (X.C.); zhangjl200101@163.com (J.Z.); rongjd@fafu.edu.cn (J.R.);; 3Experimental Center of Subtropical Forestry, Chinese Academy of Forestry, Xinyu 336600, China; cafczm1998@caf.ac.cn; 4Zhangping Forestry Bureau, Zhangping 334400, China; gq7570011@163.com (Q.C.); zpbamboo@163.com (S.L.)

**Keywords:** long-term stocking bamboo retention, moso bamboo, sustainable forest management, nutrient cycling, soil plant interactions

## Abstract

This study suggests that management of moso bamboo (*Phyllostachys edulis*) forests can increase bamboo shoot production. When farmers keep only 1- to 3-year-old bamboo stems and prevent new shoots from growing (to focus energy on producing thicker shoots), the ideal number of bamboo is 2400 plants·hm^−2^. At this density, the bamboo shoot harvest reached 18,822 kg per hectare (the highest yield). Roots became more efficient at absorbing nutrients (fine roots grew 30% longer). Soil quality improved: nitrogen and phosphorus levels increased by over 20%. Key factors driving this boost include soil moisture, organic matter, and root nitrogen content (which alone explained 75.7% of yield changes). Compared to traditional methods, this new approach increased yields by 12.5% while keeping soils healthy. This information could help bamboo farmers improve yield while protecting soil quality.

## 1. Introduction

Stand density is a critical factor in forest management, as it dramatically affects the growth conditions of forest plants, their resistance to environmental disasters, their productivity, and the nutrient balance and soil degradation of the forest soil. An appropriate stand density can optimize the growth environment for forest plants, enhance their resistance, increase their productivity, and maintain soil nutrient balance, thus preventing soil degradation [1,2]. By adjusting stand density, plant root growth and soil quality can be regulated, so as to effectively regulate forest productivity [3,4]. Kittur et al. [5] found that, under low stand density, *Dendrocalamus strictus* exhibited greater diameter at breast height and crown width, while increasing stand density led to a gradual decline in the root activity and nutrient content of *D. strictus*. Wang et al. [6] observed that the ratio of root biomass to aboveground biomass in *Larix gmelinii* plantations significantly decreased with increasing stand age and tree density. Additionally, the biomass of the underground rhizome system, internode length, specific root length, and specific root surface area of moso bamboo (*Phyllostachys edulis*) also decreased as stand density increased [7,8]. Soil quality, including its physical, chemical, and biological properties, is closely related to stand density. Studies have shown that soil porosity and pH in *Tamarix chinensis* forests reached their maximum values at moderate density, while volume weight of soil, moisture content, and electrical conductivity peaked at low density [9]. Liu et al. [10] found that in moderate-density *Robinia pseudoacacia* plantations, soil moisture, organic carbon, total nitrogen, available phosphorus, and nitrate nitrogen contents were each significantly higher than those in both low- and high-density settings. Moreover, stand density can influence the yield of *Chimonobambusa pachystachys* bamboo shoots by altering the contents of Olsen P, available nitrogen, organic matter, and available potassium in the soil [11]. These findings underscore the pivotal role of stand density in forest management; changes in stand density can alter the spatial environment, which in turn affects the soil environment and subsequently influences plant growth both directly and indirectly.

*Phyllostachys edulis* is a key economic bamboo species in Asia, and its bamboo forest ecosystem has both ecological barrier function and sustainable management value. Although the traditional intensive management mode (such as annual logging) increases the yield of bamboo in the short term, it leads to irreversible degradation of soil resources due to excessive interference: organic matter attenuation, nutrient imbalance and physical structure deterioration, which seriously restricts the long-term productivity of a bamboo forest [12,13]. In order to coordinate ecological protection and resource utilization, the long-term bamboo-stocking retention model (LR) optimizes the management of bamboo shoot forests by extending the retention period of mother bamboo (2–8 years), but the ecological threshold of its core parameter—stand density—is not yet clear. Too high density will aggravate the competition of underground rhizome roots, and too low will weaken the canopy interception function, both of which may weaken the expected benefits of the LR mode [14,15,16]. Therefore, how to scientifically quantify the optimal density range of moso bamboo forest under LR mode has become a key bottleneck to achieving sustainable management of bamboo shoot forest. Current research on the impact of stand density on bamboo forests for shoot production has primarily focused on the responses of bamboo growth and soil nutrients to stand density [17,18]. However, research on how stand density under a long-term retention model for bamboo stocking affects the growth of underground root systems and soil quality in moso bamboo forests, and consequently impacts the yield of bamboo shoots, is lacking. Therefore, the study aimed to investigate the impact of bamboo forest density on bamboo shoots, roots, and soil, identify key influencing factors, and determine the optimal management density for this management model.

An intermediate planting density is beneficial to the growth of moso bamboo and the improvement of soil quality within the long-term retention model of stocking bamboo in Zhangping City, Fujian Province, China, and how these changes affect the yield of bamboo shoots. Specifically, stand densities of 1200 ± 100, 1800 ± 100, 2400 ± 100, and 3000 ± 100 plants per hectare were established and compared with a traditional management model with a stand density of 2100 ± 100 plants per hectare. The scientific problems solved are as follows: (1) How to alleviate soil degradation caused by traditional intensive management through LR model; (2) what kind of stand density can best balance the rhizome root development–soil function synergy under LR mode; (3) how to construct a quantitative model of ‘density–root–soil–yield’ to achieve a sustainable high yield. This study provides an empirical basis for determining the optimal management density under this novel management model and it could help bamboo farmers improve yield while protecting soil quality.

## 2. Materials and Methods

### 2.1. Research Site Overview

The experimental site is situated in Shibankeng Village, Zhangping City, Longyan, Fujian Province, China (25°11′ N, 117°27′ E). The region is subject to a subtropical maritime monsoon climate, characterized by plentiful hydrothermal resources and a pronounced distinction between wet and dry seasons. The average temperature is 19.2 °C, and the frost-free period lasts between 289 and 303 days, with an annual average precipitation accumulation of 1450 mm. The topography is primarily composed of low mountains, with hills as a secondary feature, and the site is situated at approximately 565 m above sea level. The predominant soil type is red-yellow soil, with a soil profile depth exceeding 100 cm. The principal vegetation comprises *Crassocephalum crepidioides*, *Oxalis corniculata*, *Lophatherum gracile*, and *Microlepia hancei*. The total forested area covers 1520 hm^2^, with bamboo forest spanning 400 hm^2^. In Shibankeng Village, the bamboo forests exhibit lower productivity in odd-numbered years, higher productivity in even-numbered years, and larger yields in even-numbered years, without adjustment for masting years. The soil data from the previous measurements are as follows: average moisture content, 12.76%; pH, 4.3; soil organic matter, 38.96 g·kg^−1^; total nitrogen, 0.81 g·kg^−1^; total phosphorus, 0.26 g·kg^−1^; total potassium, 12.35 g·kg^−1^; available phosphorus, 4.97 mg·kg^−1^; available potassium, 49.62 mg·kg^−1^.

### 2.2. Research Site Setup

In October 2019, within the bamboo shoot cultivation area in Shibankeng Village, Zhangping City, Longyan, Fujian Province, China, an area of bamboo forest was established for experimental purposes under a novel management paradigm. The selection criteria included consistent growth and management practices (encompassing annual weeding and soil aeration at least once), uniform water and nutrient regimes (irrigation applied following 20 consecutive rain-free days and fertilization conducted quarterly), homogenous site conditions (i.e., consistent elevation, slope aspect, slope position, and gradient), and diverse stand densities. The bamboo forests were established under a long-term management paradigm with varying stand densities: D1, 1200 ± 100 plants·hm^−2^; D2, 1800 ± 100 plants·hm^−2^; D3, 2400 ± 100 plants·hm^−2^; and D4, 3000 ± 100 plants·hm^−2^. The control group (CK) comprised bamboo forests managed under conventional shoot production practices, including standard selective thinning. In line with the methodologies of Cai et al. [13] and Li et al. [19], the initial retention of new bamboo in the CK group was set at 900 plants·hm^−2^, with the removal of bamboo older than seven years, resulting in a density of approximately 2100 ± 100 plants·hm^−2^. The age structure of the bamboo forest was stratified according to a ratio of 1:2:2:1 for classes I, II, III, and IV, respectively. Each treatment and the control group (CK) were replicated three times, totaling 15 plots, each measuring 25.8 m × 25.8 m. A buffer zone exceeding 3 m was established between each plot in which bamboo and associated rhizomes were eliminated, thereby mitigating interference among treatments.

### 2.3. Test Methods

#### 2.3.1. Collection and Processing of Root Samples

In March, June, September, and December of 2022, three representative bamboo (*Phyllostachys edulis*) plants were randomly selected from each experimental site as standard bamboo culms. Soil samples were collected within a 60 cm radius from the base of each reference bamboo culm, oriented according to the four cardinal directions (north, south, east, and west). Surface debris was removed, and soil cores were extracted from each sampling point using a 65 mm diameter soil auger (Shaoxing Bowei Instrument Co., Ltd., Zhejiang, China) and stratified into three depth intervals: 0–20 cm, 20–40 cm, and 40–60 cm [14,20]. Roots from the four quadrants within the same stratum were pooled, sealed in labeled plastic bags, and transported to the laboratory for further analysis.

Roots were rinsed using a 0.5 mm aperture soil sieve, and the cleaned roots were deposited into an evaporation dish. Employing tweezers, a magnifying glass, and calipers (Shanghai Xiusong Co., Ltd., Shanghai, China), roots ≤ 2 mm in diameter were separated. Live and dead roots were differentiated based on their morphological and color characteristics and categorized into two diameter classes: 0–1 mm and 1–2 mm. After sorting, the roots were digitized using a scanner (EPSON Perfection V700 Photo, Epson, Suwa, Japan), and morphological parameters, including root length (cm), average root diameter (cm), surface area (cm^2^), and volume (cm^3^), were quantified using specialized software (WinRHIZO Pro 2009b; Regent Instruments, Québec City, QC, Canada). Based on these data, specific root length (cm·g^−1^), root tissue density (g·cm^−3^), and specific root surface area (cm^2^·g^−1^) were computed, according to the following equations:(1)$$\mathrm{Specific}\;\mathrm{Root}\;\mathrm{Length},\mathrm{SRL}=\frac{\mathrm{Root}\;\mathrm{Length}\;(\mathrm{cm})}{\mathrm{Root}\;\mathrm{Dry}\;\mathrm{Weight}\;(g)}$$(2)$$\mathrm{Root}\;\mathrm{Tissue}\;\mathrm{Density},\mathrm{RTD}=\frac{\mathrm{Root}\;\mathrm{Dry}\;\mathrm{Weight}\;(g)}{{\mathrm{Volume}\;(\mathrm{cm}}^{3})}$$(3)$$\mathrm{Specific}\;\mathrm{Root}\;\mathrm{Surface}\;\mathrm{Area},\mathrm{SRA}=\frac{{\mathrm{Root}\;\mathrm{Surface}\;\mathrm{Area}\;(\mathrm{cm}}^{2})}{\mathrm{Root}\;\mathrm{Dry}\;\mathrm{Weight}\;(g)}$$

The roots from each layer were homogenized and heated in an oven (Shanghai Boxun Industry Co., Ltd., Shanghai, China) at 105 °C for 30 min to deactivate their enzymes. Subsequently, they were dried to a constant mass at 80 °C, after which their weights were determined to an accuracy of 0.001 g. The roots were then ground using a grinder (DC-500A model, Zhejiang Wuyi Dingzang Daily Metal Products Factory, Jinhua, China) and passed through a 0.149 mm sieve before being bagged for nutrient content analysis of the bamboo roots.

Total nitrogen content was quantified using the semi-micro Kjeldahl method (LY/T 1270-1999 [21]. Instrument used: Kjeldahl nitrogen analyzer, Lichen Technology Instrument Co., Ltd., Shanghai, China). Total phosphorus content was assayed using the molybdenum antimony colorimetric method (LY/T 1271-1999 [22]. Instrument used: UV spectrophotometer, Jinghua Technology Instrument Co., Ltd., Shanghai, China). Total potassium content was assessed via the ammonium acetate leaching-flame photometer method (LY/T 1270-1999 [22]. Instrument used: Atomic Absorption Spectrophotometer, Jian Shi (Beijing) Analytical Instrument Co., Ltd., Beijing, China). Organic carbon content was evaluated using the potassium dichromate–external heating method (LY/T 1237-1999 [23]. Instrument used: titrator, Jianhu Huabo Glass Products Co., Ltd., Yancheng, China).

#### 2.3.2. Collection and Processing of Soil Samples

In March, June, September, and December of 2022, soil samples were collected from the sampling site using the S-shaped curve random sampling method. A ring knife (Cangzhou Kanxin Instrument Co., Ltd., Changzhou, China) was used to collect soil from depths of 0–20, 20–40, and 40–60 cm for physical property analysis. Approximately 300 g of soil from each layer was placed into sealed bags and brought back to the laboratory for nutrient analysis.

Soil physical properties were measured according to the methods described in LY/T 1215-1999 [24] (main instruments used: analytical balance, Ohaus Instruments Shanghai Co., Ltd., Shanghai, China; oven, Shanghai Boxun Industry Co., Ltd., Shanghai, China; dryer, Xiangbo Glass Co., Ltd., Zhuzhou, China). For the analysis of soil chemical properties, the samples underwent air-drying, followed by subsequent oven-drying, grinding, and sieving through 0.149 mm mesh prior to storage.

The PH value was measured by a PH tester (Shanghai Yidian Scientific Instrument Co., Ltd., Shanghai, China); the conductivity was measured by a conductivity meter (Shanghai Yidian Scientific Instrument Co., Ltd., Shanghai, China). The content of total nitrogen was quantified using the Kjeldahl method (LY/T 1228-1999 [25]. Instrument used: Kjeldahl nitrogen analyzer, Lichen Technology Instrument Co., Ltd., Shanghai, China). Total phosphorus content was ascertained using the molybdenum-antimony colorimetric method (LY/T 1232-1999 [26]. Instrument used: UV spectrophotometer, Shanghai Jinghua Scientific Instrument Co., Ltd., Shanghai, China). Total potassium levels were evaluated by atomic absorption spectrophotometry (LY/T 1234-1999 [27]. Instrument used: Atomic Absorption Spectrophotometer, Jian Shi (Beijing) Analytical Instrument Co., Ltd., Beijing, China). Soil organic matter content was assessed using the potassium dichromate-heating method (LY/T 1237-1999 [23]. Instrument used: titrator, Jianhu Huabo Glass Products Co., Ltd., Yancheng, China). Available potassium was assessed using the ammonium acetate extraction–flame photometry method (LY/T 1236-1999 [28]. Instrument used: Atomic Absorption Spectrophotometer, Jian Shi (Beijing) Analytical Instrument Co., Ltd., Beijing, China). Furthermore, available phosphorus content was quantified via hydrochloric acid–sulfuric acid extraction (LY/T 1233-1999 [29]. Instrument used: UV spectrophotometer, Shanghai Jinghua Scientific Instrument Co., Ltd., Shanghai, China).

#### 2.3.3. Bamboo Shoot Yield Determination

From 1 March to 30 April 2022, during the bamboo shoot development period, samples of bamboo shoots exceeding 30 cm in height above the ground were harvested from each plot every two days. The fresh shoots, still encased in their culms, were weighed in situ using an analytical balance (Ohaus Instruments Shanghai Co., Ltd., Shanghai, China) to obtain data on shoot yield.

### 2.4. Data Analysis

Data processing and statistical analysis were performed utilizing Excel 2019 (Microsoft Corp., Redmond, WA, USA) and SPSS 25.0 (IBM Corp., Armonk, NY, USA) software. Shapiro–Wilk test was used to test the normality of data (*p* > 0.05), Levene’s test was used to test the homogeneity of variance (*p* > 0.05), and the data that did not meet the conditions were retested after logarithmic (log_10_) or square root conversion [30]. One-way ANOVA followed by the least significant difference (LSD) multiple comparison method was employed to determine differences between groups, at a significance threshold of *p* < 0.05 [31]. Experimental data are expressed as mean ± standard deviation values. Principal component analysis was utilized for factor selection and comprehensive evaluation. Heatmap analysis and plotting were conducted using Origin 2024 software (OriginLab, Northampton, MA, USA), and Pearson correlation analysis was utilized to assess the relationship between the morphological characteristics of bamboo rhizomes and nutrient content. Additionally, redundancy analysis and plotting of bamboo growth traits in relation to soil nutrients were conducted using Canoco 5.0 software (Microcomputer Power, Ithaca, NY, USA).

## 3. Results

### 3.1. Impact of Stand Density on Bamboo Shoot Yield and Quantity

As depicted in Figure 1, under the novel management treatments, bamboo shoot yield and number exhibited a unimodal response to stand density, reaching their peak under density D3 (18,822 kg·hm^−2^ and 7080 shoots·hm^−2^, respectively). The bamboo shoot yields under the D1, D2, D3, and D4 treatments significantly increased, by 21.6%, 45.1%, 52.8%, and 22.1%, respectively, compared to the control group (CK) (*p* < 0.05). Additionally, the number of shoots at densities D3 and D4 significantly increased, by 11.2% and 12.5%, respectively, compared to CK conditions (*p* < 0.05).

### 3.2. Impact of Stand Density on Bamboo Root Growth Traits

#### 3.2.1. Key Root Morphological Traits Under Different Stand Densities

As depicted in Figure 2, under the novel management treatments, the specific root length and root tissue density of roots in the 0–1 mm size class also exhibited a unimodal response to stand density. The specific root length under density D3 reached its maximum value (9.84 cm·g^−1^), corresponding to a significant increase of 29.8% compared to the control group (CK) (*p* < 0.05), while no significant differences in root tissue density were observed among the treatment groups (*p* > 0.05). The specific root surface area also reached its peak at density D3 (6.25 cm·g^−1^), significantly increasing by 24.5% compared to CK conditions (*p* < 0.05).

In the new management model, the changes in tissue density of the 1–2 mm size class roots followed a pattern consistent with that of the 0–1 mm size class. Density D2 exhibited the highest root tissue density (0.16 g·cm^−3^), which was significantly higher, by 23.1%, compared to CK conditions (*p* < 0.05). The specific root length under the four new management models was lower than that under CK conditions, with density D1 showing a significant decrease of 45.3% compared to CK conditions (*p* < 0.05). No significant differences in specific root surface area were found among the novel management treatment groups (*p* > 0.05).

#### 3.2.2. Nutrient Content of Bamboo Roots

As depicted in Figure 3, the total nitrogen, total phosphorus, and total potassium contents of the 0–1 mm roots of moso bamboo were highest under density D3 (10.2 g·kg^−1^, 1.6 g·kg^−1^, and 5.0 g·kg^−1^, respectively). Notably, the total nitrogen content under density D3 was significantly increased by 12.9% compared to CK conditions (*p* < 0.05). Under the novel management treatments, the organic carbon content of moso bamboo roots exhibited a unimodal response to stand density, although there were no significant differences in organic carbon content among the treatment groups (*p* > 0.05).

In the context of the novel management model treatments, the organic carbon, total nitrogen, and total potassium contents of the 1–2 mm roots of moso bamboo also initially exhibited a unimodal response to stand density, while total phosphorus instead exhibited a consistently increasing trend. The organic carbon and total phosphorus contents of the bamboo roots were highest under density D4 (462.1 g·kg^−1^ and 2.1 g·kg^−1^, respectively). The total phosphorus contents under the four new management treatments significantly increased, by 39.4%, 72.7%, 103.0%, and 111.1%, compared to CK conditions (*p* < 0.05). The total nitrogen and total potassium contents of the bamboo roots were highest under density D3 (8.3 g·kg^−1^ and 5.0 g·kg^−1^, respectively). The total nitrogen contents under the D1, D2, D3, and D4 treatments exhibited significant increases, of 15.4%, 16.4%, 30.2%, and 11.8%, respectively, compared to CK conditions (*p* < 0.05). Additionally, the total potassium content under density D3 was significantly higher than that under CK conditions, by 16.1% (*p* < 0.05), whereas the total potassium content under density D1 was significantly lower than that under CK conditions, by 12.4% (*p* < 0.05).

### 3.3. Impact of Stand Density on Bamboo Forest Soil Quality

#### 3.3.1. Physical Properties of Bamboo Forest Soil

As shown in Figure 4, in the 0–20 cm soil layer, under the new management model, the soil capillary porosity and total porosity increased with stand density. Specifically, the capillary porosity under densities D1, D2, D3, and D4 was significantly reduced, by 23.6%, 18.2%, and 16.6%, respectively, compared to CK conditions (*p* < 0.05), while the total porosity under D1 and D2 densities decreased significantly, by 13.2% and 11.2%, respectively, compared to CK conditions (*p* < 0.05). The volume weight of soil under D1 was the highest (1.2 g·cm^−3^), which represented a significant increase of 12.6% compared to CK conditions (*p* < 0.05). There were no significant differences in soil moisture content or non-capillary porosity among the treatment groups (*p* > 0.05).

In the 20–40 cm soil layer, the soil moisture content, total porosity and bulk density under the new management model exhibited a unimodal response to stand density. The soil moisture content under the D1, D2, D3, and D4 treatments significantly increased, by 13.2%, 14.8%, 17.7%, and 16.3%, respectively, compared to CK conditions (*p* < 0.05). The bulk density under D1 was the highest (1.3 g·cm^−3^), but the total porosity at this density was significantly lower than that under CK conditions, by 14.0% (*p* < 0.05). The capillary porosity under D1 and D3 densities was significantly reduced, by 24.1% and 18.6%, respectively, compared to CK conditions (*p* < 0.05). There were no significant differences in non-capillary porosity among the treatment groups (*p* > 0.05).

In the 40–60 cm soil layer, the non-capillary porosity under density D1 was the lowest (8.2%), corresponding to a significant decrease of 23.5% compared to CK conditions (*p* < 0.05). There were no significant differences in soil moisture content, bulk density, capillary porosity, or total porosity among the treatment groups (*p* > 0.05).

#### 3.3.2. Determination of Basic Chemical Properties of Soil in Moso Bamboo Forest

In the 0–20 cm soil layer, with increasing stand density, the contents of soil organic matter, total nitrogen, total potassium, total phosphorus, available phosphorus, and readily available potassium under the new management model treatments all exhibited a unimodal response to stand density. In contrast, the soil pH value exhibited a continuously increasing trend (Figure 5). The highest levels of total nitrogen, available phosphorus, readily available potassium, and electrical conductivity were observed under density D3 (1.9 g·kg^−1^, 9.9 mg·kg^−1^, 121.0 mg·kg^−1^, and 0.2 ms·cm^−1^, respectively), with increases of 13.3%, 28.1%, and 111.1% (*p* < 0.05), respectively, compared to the control (CK) conditions. However, the soil organic matter content under D3 was significantly lower than that under CK conditions, by 14.8% (*p* < 0.05). The highest contents of total potassium and total phosphorus were recorded under density D1 (13.8 g·kg^−1^ and 0.4 g·kg^−1^, respectively), with total phosphorus under D1 significantly increased by 76.9% compared to CK conditions (*p* < 0.05). The soil pH value was highest under density D4 (4.6). The C/N ratio was the highest in CK in all soil layers.

In the 20–40 cm soil layer, as stand density increased, the contents of soil organic matter, total phosphorus, total potassium, available phosphorus, quick-acting potassium, and pH value also exhibited a unimodal response to stand density. The highest levels of soil organic matter, available phosphorus, quick-acting potassium, pH value, and electrical conductivity were observed under density D3 (53.2 g·kg^−1^, 2.9 mg·kg^−1^, 107.8 mg·kg^−1^, and 0.1 ms·cm^−1^, respectively), with significant increases of 24.3%, 12.1%, and 75.0% (*p* < 0.05), respectively, in available phosphorus, readily available potassium, and electrical conductivity compared to CK conditions. The total potassium content under each of the four new management model treatments was significantly lower than that under CK conditions. The highest total nitrogen and total phosphorus contents were found under density D4 (1.0 g·kg^−1^ and 0.2 g·kg^−1^, respectively), with significant increases of 21.4% and 23.8%, respectively, compared to CK conditions (*p* < 0.05).

In the 40–60 cm soil layer, the changes in soil organic matter, total potassium, total nitrogen, total phosphorus, and available phosphorus under the new management model were consistent with those observed in the 0–20 cm soil layer, while the soil pH value also exhibited a unimodal response to stand density. The highest contents of total nitrogen, available phosphorus, quick-acting potassium, pH value, and electrical conductivity were observed under density D3 (1.4 g·kg^−1^, 1.1 mg·kg^−1^, 101.5 mg·kg^−1^, 4.4, and 0.1 ms·cm^−1^, respectively). Under D3 density, total nitrogen, available phosphorus, quick-acting potassium, and electrical conductivity significantly increased by 32.6%, 16.5%, 14.9%, and 22.2% (*p* < 0.05), respectively, compared to the control (CK). The total phosphorus content under the D1, D2, D3, and D4 treatments significantly increased by 179.1%, 134.9%, 109.3%, and 129.1% (*p* < 0.05), respectively, compared to CK conditions, while the soil organic matter content decreased significantly by 42.3%, 34.4%, 20.0%, and 22.3% (*p* < 0.05), respectively.

There were significant differences in the chemical properties of the bamboo forest soil across the different soil layers (*p* < 0.05). In various stand density treatments, the levels of organic matter, total nitrogen, total phosphorus, available phosphorus, readily available potassium, and electrical conductivity in the 0–20 cm soil layer were significantly higher than those in the other two layers (*p* < 0.05). The total nitrogen, available phosphorus, readily available potassium, and electrical conductivity in each soil layer reached their highest values under D3 density. The C/N ratio in the 20–40 cm soil layer was significantly higher than that in the other two soil layers (*p* < 0.05). The pH value of the soil under D3 density showed no significant differences among the various soil layers (*p* > 0.05), while the highest pH value under other stand density treatments was observed in the 0–20 cm soil layer. However, the total potassium content in the 0–20 cm soil layer was lower than that in the other two soil layers across all stand density treatments (*p* < 0.05).

### 3.4. Relationships Among Root Growth Traits, Soil Quality, and Bamboo Shoot Yield and Quality

In the redundancy analysis presented in Figure 6, bamboo shoot yield and quantity were response variables, and root growth traits and soil quality were explanatory variables. The first ordination axis explained 94.59% of the total variance, while the second axis explained the remaining 5.41%. The total nitrogen content in the roots had a significant impact on shoot yield and quantity, accounting for 75.7% of the variance. This was followed by the specific root surface area of 1–2 mm specific root surface area (14.7%), volume weight of soil (5.1%), total potassium content in the roots (4.3%), and capillary porosity (0.2%), with the remaining indicators having a minimal effect. There was a strong positive correlation between both bamboo shoot yield and quantity and total nitrogen and total potassium content in the roots, while a strong negative correlation was observed for both bamboo shoot yield and quantity and the specific root surface area of 1–2 mm roots, volume weight of soil, and capillary porosity.

### 3.5. Relationships Between Bamboo Growth Traits and Soil Quality Indices

As shown in Figure 7, Pearson correlation analysis indicated that 34 correlations between bamboo growth traits and soil quality indices were highly significant (*p* < 0.01), while another 18 pairs of variables also exhibited significant correlations (*p* < 0.05). This suggests varying degrees of correlation between the growth traits of bamboo and soil quality indices across the different stand densities. Soil moisture content showed an extremely significant positive correlation with bamboo shoot yield, shoot quantity, the specific root length of 0–1 mm roots, and the nitrogen and phosphorus content in the roots. It also displayed a significant positive correlation with the specific root surface area of 0–1 mm roots and the potassium content of the roots. However, total potassium content in soil was found to have an extremely significant negative correlation with the aforementioned indicators. The growth traits of bamboo exhibited both positive and negative correlations with soil quality indices to varying degrees, indicating a mutual dependence and likely interaction between bamboo and soil during growth.

### 3.6. Principal Component Analysis of Factors Influencing Bamboo Shoot Production Under Different Stand Densities

Principal component analysis (PCA) was used to comprehensively evaluate the growth potential of bamboo shoots. The data were standardized by Z-score, and the applicability of the data was verified by Kaiser–Meyer–Olkin test and Bartlett spherical test. According to the eigenvalue > 1 and the cumulative contribution rate ≥ 85%, five principal components were extracted (cumulative 100%, Table 1). PC1 (41.16 %): soil total phosphorus (0.92), yield of moso bamboo shoots (0.95), 0–1 mm root total nitrogen (0.91), characterization of ‘phosphorus–nitrogen synergistically driven productivity core’: soil phosphorus pool supports root nitrogen absorption and jointly promotes bamboo shoot yield. PC2 (27.38%): soil organic carbon (0.96), 1–2 mm root length (0.92), reflecting ‘carbon storage–transport root development coupling’, high-organic carbon soil promotes coarse root growth and enhances nutrient transport capacity. PC3 (17.62%): 0–1 mm root tissue density (−0.91), 1–2 mm root tissue density (−0.86), and root surface area (0.83), revealing the ‘root function trade-off’. High-surface area absorbing roots (positive load) and low tissue density (negative load) show that resources are preferentially invested in absorption rather than structural reinforcement. The stand density treatments can be ranked in order descending order of their growth condition scores as follows: D3, D4, D2, CK, D1 (Table 2).

## 4. Discussion

### 4.1. Bamboo Shoots

Density regulation in bamboo forests is a key management practice for optimizing bamboo shoot production [32]. Our research determined that bamboo shoot yield and number exhibited a unimodal response to stand density, reaching their maximum values under density level D3. Furthermore, this analysis suggests that the growth of bamboo shoots is closely related to soil conditions and fine root systems. As bamboo forest density increases, competition among trees intensifies, leading bamboo to absorb more water and nutrients from the soil by increasing their root biomass to support shoot growth [33]. However, excessively high forest density can significantly impact competition for soil resources among root systems. In contrast, when forest density becomes too high, the amount of light energy and the duration of sunlight received per unit area of bamboo leaves can decrease, which hinders photosynthesis, thus reducing the biosynthesis of organic matter and subsequently decreasing the transport of organic materials to the root systems, thereby impairing bamboo shoot growth [34]. Thus, forest density not only reflects competition among mature bamboo trees but also directly influences light conditions within the forest, which in turn affect the yield and number of bamboo shoots [35]. D3 is the optimal density range, which provides a key threshold for scientific thinning and structural adjustment of the bamboo shoot forest, and helps operators avoid the loss of yield caused by blind pursuit of high density or low density. Compared with the traditional management mode, the shoot yields and shoot numbers of moso bamboo with D1–D4 density under the new management mode were higher than those under the traditional management mode. It can be seen that the long-term bamboo-stocking retention model may not only improve the yield of bamboo shoots and expand the economic benefits, but also optimize the age structure and spatial allocation of bamboo forests, to more effectively coordinate the above-ground photosynthetic capacity and the absorption capacity of underground resources, thus breaking through the bottleneck of resource utilization that may exist under the traditional model. In the future, the adaptability of this model under different site conditions can be further studied, and its long-term ecological benefits can be evaluated to optimize the comprehensive management strategy.

### 4.2. Bamboo Root Growth Traits

The morphology of fine roots is intricately linked to their functional capacity, determining how effectively plants are able to utilize soil resources. Changes in fine root morphological traits can reflect shifts in their functional capabilities [36]. Our research found that at density level D3, the fine roots (0–1 mm diameter) of moso bamboo exhibited the highest specific root length (SRL)and specific root surface area (SRSA), while the 1–2 mm root tissue density (RTD) under density level D2 had the greatest tissue density. It is well understood that the nutrients enabling plant root growth primarily come from the soil. As density increases, the amount of litter per unit area rises, which enhances ground cover and mitigates the erosive impact of rainfall [37]. This, in turn, contributes to water conservation, improves soil structure, promotes microbial activity, and boosts soil fertility [38]. Under the input of unit root biomass, high SRL and SRSA were conducive to the construction of a longer and larger surface area absorption root network in the moso bamboo, which directly enhanced the contact interface between root and soil, and significantly improved the efficiency of root absorption of water and mineral nutrients [39,40]. This morphological adjustment is an active physiological adaptation strategy for moso bamboo to cope with the competition pressure of underground resources caused by the increase in stand density. It aims to maintain or promote the supply of nutrients required for bamboo shoot growth by improving the ‘foraging’ ability of resources per unit root biomass. On the other hand, the RTD of 1–2 mm roots was the largest under D2 density, which may reflect the strengthening of root structure support and resource transport function. Increased RTD usually means that the root tissue is denser, the cell wall is thicker, and the degree of lignification is higher, which enhances the mechanical strength of the root and the long-distance transport efficiency of water/nutrients [41,42]. It helps to transport the absorbed resources more effectively to the ground for the rapid growth of bamboo shoots. Therefore, the change in stand density drives the functional trade-off and plasticity adjustment of fine roots between absorption efficiency type (high SRL/SSA) and transport/support efficiency type (high RTD) by affecting soil microenvironment (water, nutrient availability, microbial activity) and intraspecific competition intensity. This is the embodiment of key physiological processes such as resource acquisition, distribution and transportation of moso bamboo roots to adapt to environmental changes. Conversely, the tissue density of the 1–2 mm diameter roots exceeded that of the 0–1 mm diameter roots, possibly as a result of the strong complementary effects between root length density and specific root length occurring with changes in bamboo forest density [43].

Carbon, nitrogen, phosphorus, and potassium are the fundamental nutrients required for plant growth and are closely related to the synthesis of substances and physiological metabolism within a plant [44]. In our study, the total nitrogen, phosphorus, and potassium contents in the 0–1 mm diameter roots of D3 density moso bamboo were found to be the highest. In the context of the novel management model treatments, the organic carbon, total nitrogen, and total potassium content in the 1–2 mm diameter roots of moso bamboo exhibited a unimodal response to stand density, while total phosphorus consistently increased. Additionally, total nitrogen and total potassium content were highest under D3 density, whereas organic carbon and total phosphorus content peaked under D4 density. Numerous studies have indicated that an increase in stand density can enhance the accumulation of litter in the soil, thereby improving soil quality to support the growth of moso bamboo roots. Furthermore, moso bamboo roots possess relatively high specific root lengths and specific root surface areas, conferring their relatively high resource foraging efficiency [45,46]. Thus, the nutrient content of moso bamboo roots tended to increase with stand density. However, excessively high stand densities can adversely affect root system growth. In this study, the absorption capacity of the 0–1 mm diameter roots was greater than that of the 1–2 mm diameter roots, likely owing to the more active physiological metabolic activity levels of smaller diameter roots, which have a higher rate of cell division at their root tips [47,48]. Thus, under a long-term management model of retaining the stocking of bamboo plants, an appropriate stand density can enhance the morphological structure of moso bamboo roots and enhance their nutrient absorption capacity.

Our study found that the nitrogen content of roots (especially 0–1 mm roots) reached the peak at D3 density, which was highly consistent with the phenomenon that the yield of bamboo shoots was the highest at D3 density, indicating that there was a close physiological relationship between root nitrogen storage/supply capacity and aboveground productivity. Nitrogen is the main component of protein, nucleic acid, chlorophyll and other key life substances, which is essential for cell division, tissue differentiation and the rapid expansion of bamboo shoots [49]. We believe that the optimal soil environment and efficient root morphology under D3 density jointly promote the efficient absorption of soil nitrogen by roots. Some of the absorbed nitrogen may be directly assimilated into organic nitrogen compounds such as amino acids and stored in the roots, while the other part is rapidly transported upward through the xylem to the developing bamboo shoots [50,51]. During the peak period of spring shooting, the nitrogen stored in the root system can be quickly mobilized and transported to the shoot tip, providing sufficient nitrogen guarantee for the explosive growth of the shoot [52]. Therefore, the maximization of root nitrogen content under D3 density is likely to directly contribute to the highest yield of bamboo shoots by ensuring the nitrogen supply during the critical period of the rapid growth of bamboo shoots. This finding indicates the central role of underground nitrogen uptake, storage and transport efficiency in regulating the aboveground productivity of moso bamboo. Future studies can further quantify the quantitative relationship between root nitrogen pool dynamics and bamboo shoot growth rate and nitrogen demand.

In summary, under the long-term mother bamboo conservation mode, reasonable stand density significantly improved the acquisition efficiency of soil resources (especially nitrogen) and its own nutrient storage level by optimizing the soil environment and inducing favorable morphological plasticity of roots. The optimization of these underground processes, especially the enhancement of root nitrogen absorption and storage capacity, has laid a key physiological foundation for the high yield of bamboo shoots.

### 4.3. Bamboo Forest Soil Quality

Soil is a principal site of biological activity in forest ecosystems, providing essential nutrients for plant growth and serving as a pivotal nexus for the transformation of nutrient elements. The level of soil productivity directly determines the biological yield and functions of a forest ecosystem [53]. Our study found minimal differences in soil moisture content, bulk density, and non-capillary porosity in the 0–60 cm soil layer across the different stand density treatments. This is likely because the study sites with varying stand densities were all located in the same region, where local factors, such as rainfall and topography, exert relatively uniform influences on soil moisture, bulk density, and non-capillary porosity [54,55,56]. Consequently, discrepancies in these soil attributes among the different stand densities were negligible. In the 0–40 cm soil layer, capillary porosity and total porosity were greatest in the control group (CK); however, within the 40–60 cm layer, no significant differences in these porosity indices were observed among the treatment groups. This pattern may stem from the sparse root distribution associated with low stand densities leading to a reduction in soil pore space and compromised water retention capabilities. Conversely, at excessively high densities, increased competition among trees can hinder soil development and plant growth, consistent with both excessively high and low stand densities being detrimental to soil structure improvement [57]. Research by Lan et al. [58] has shown that pore conditions are influenced not only by stand density but also by stand spatial structure, litter layer, soil fauna, and microbial activity. Thus, it can be inferred that a bamboo stand density of approximately (2100 ± 100) plants·hm^−2^ is optimal for enhancing soil water retention capacity and improving soil structure. Additionally, soil porosity tends to decrease with soil depth. Therefore, when managing moso bamboo forests, it is important to strengthen soil management practices, implement reasonable soil tillage and loosening, remove old and dead culms, reduce soil compaction, and promote the growth of fibrous roots, thereby increasing bamboo yields.

Soil chemical properties are important indicators of soil fertility, directly influencing plant growth and development [59]. The principal soil chemical properties include pH and the contents of organic matter, nitrogen, phosphorus, potassium, and additional elements. The combined effects of these factors can reflect the overall chemical quality of the soil [60]. In our study, the change in soil C/N ratio provides important information on soil organic matter decomposition and nitrogen cycling. The results showed that the C/N ratio in each soil layer was the highest in CK treatment, indicating that the decomposition of soil organic matter may be relatively slow under low stand density. A high C/N ratio usually means that soil nitrogen is relatively deficient or organic matter is more complex and difficult to decompose, which may limit the mineralization and decomposition rate of organic matter by microorganisms and the utilization efficiency of nitrogen by plants [61,62]. Within our study, the soil pH across the various density treatments was strongly acidic. The 0–20 cm soil stratum exhibited the highest pH values under density level D4, whereas the 20–60 cm stratum exhibited the highest pH values under density level D3. This finding may be owing to low stand density leading to excessive light and heat, thus disrupting the moisture environment beneath the forest floor, resulting in relatively dry soil. Increased weathering and leaching can cause the loss of base cations, facilitating the development of acidic soils [63]. Our study found that the pH value of surface soil under different stand density treatments was higher than that of deeper soil layers. This finding can possibly be attributed to the rapid decomposition of surface litter generating an abundance of acidic substances that accumulate in the upper soil layers, where warmer conditions and more light tend to have an impact on chemical properties [64].

Studies have demonstrated that stand density significantly influences the physical and chemical properties of soil, particularly soil organic matter and total porosity. At an optimal stand density, higher soil nutrient content is maintained, which is conducive to sustaining a healthy forest structure and favorable soil physicochemical properties [65]. For example, there are significant differences in the soil chemical properties of *Pinus armandii* plantations under varying densities; specifically, the content of soil organic matter decreased with stand density, and both excessively high and low pH values were found to be detrimental to the accumulation of organic matter and its nutrient elements [66]. These findings underscore the importance of stand density as a key factor shaping soil chemistry in forest ecosystems. Our present results indicate that bamboo forest stand densities of approximately 2100–2400 ± 100 culms·hm^−2^ yield the highest contents of soil organic matter, total nitrogen, available phosphorus, and readily available potassium. This may be attributed to the moderate light, moisture, and temperature conditions within this stand density range, which likely facilitated the proliferation and metabolic activities of soil microorganisms. Suitable temperatures enhance microbial enzyme activity, accelerating the decomposition of organic matter and nutrient cycling [67,68]. In the present study, under medium and high stand densities, the contents of total phosphorus and total potassium in the soil were lower. In previous work, the growth of moso bamboo under medium and high stand densities exhibited a higher demand for phosphorus and potassium, which increased the root system’s absorption of these nutrients [69]. Conversely, total potassium content in the soil was highest at a depth of 20–60 cm under the control conditions (CK), likely owing to the more favorable environment for microbial decomposition of organic matter at this density, which in turn elevated total potassium levels in the soil. This suggests that, beyond management practices, stand density is the primary factor influencing total potassium content in the soil. Chen et al. [70] found that the soil electrical conductivity of *Tamarix chinensis* at medium and high stand densities was lower than that at low density, with conductivity gradually decreasing as stand density increased. However, in our study, the electrical conductivity of moso bamboo at medium and high densities was higher than that at low density. This discrepancy may be owing to the lower canopy closure that occurs at low stand densities, resulting in reduced interception of rainwater, which can severely erode the soil and wash away salts, thereby lowering soil electrical conductivity. As canopy closure increases with higher stand densities, rainwater erosion is mitigated, leading to a gradual increase in electrical conductivity [71,72,73].

The enzyme activity is relatively high in the shallow soil of moso bamboo forests, as these soils are characterized by abundant organic matter and minerals, as well as appropriate moisture and air content. These factors collectively contribute to the higher distribution of total nitrogen, total phosphorus, total potassium, available phosphorus, quickly available potassium, and organic matter observed in the shallow soil layer. Moreover, there was a notable presence of major base cations, such as Ca^2+^, Mg^2+^, and K^+^, which enhance the soil’s electrical conductivity. Additionally, the pH value of the shallow soil aligns more favorably with the growth of moso bamboo [38,74,75,76,77,78].

Our study revealed that under different forest density treatments, the contents of organic matter, total nitrogen, total phosphorus, available phosphorus, and readily available potassium, as well as electrical conductivity, in the 0–20 cm soil layer were higher than those in the other layers. Other than in the D3 density treatment, the soil pH value also reached its maximum in the 0–20 cm layer. Conversely, the total potassium content in the 0–20 cm layer was significantly lower under all forest density treatments compared to the other two layers, likely owing to the leaching of K^+^ ions from the surface soil caused by water percolation [79]. The content of organic matter and nutrients in the 0–20 cm soil layer was high, but its C/N ratio was significantly lower than that in the 20–40 cm soil layer. This further confirms that the decomposition of organic matter in the surface soil is relatively active (lower C/N ratio), and the nutrient turnover is faster, but it is also accompanied by the risk of leaching of potassium and other elements [80]. The higher C/N ratio in the 20–40 cm soil layer indicates the stability of organic matter decomposition and the relative limitation of nitrogen in this layer [81].

These findings underscore the importance of proper forest density management in influencing soil quality within moso bamboo forests under a regime of long-term bamboo shoot cultivation. Consequently, adjusting soil management strategies in bamboo forests is crucial for fostering the growth and development of root systems, thereby attaining productive management objectives.

### 4.4. Correlations

The root system serves as the principal organ for plant water and nutrient uptake, with its development exerting an integral influence on plant growth. Nitrogen, a nutrient element critical to plant development, plays a crucial role in processes such as protein formation, photosynthesis, and cell division [82]. In our study, the roots’ total nitrogen content emerged as a predominant factor affecting bamboo shoot yield and production numbers. However, soils with excessively high bulk density and capillary porosity can lead to declines in soil fertility, causing land productivity to deteriorate. There was a strong negative correlation between bamboo shoot yield and both volume weight of soil and capillary porosity. Plant root systems facilitate the cycling of nutrients in the soil and the formation of aggregates, playing a vital role in soil stabilization and serving as a source of soil organic matter. In turn, the soil provides the water and nutrients necessary for plant growth, along with a suitable growth environment [83]. Thus, there is a mutual dependency and interaction between bamboo and the soil throughout the growth process. The present study employed a principal component analysis for a comprehensive evaluation of the various stand density treatments, revealing that the growth condition of bamboo forests for shoot production was optimal under D3 density, while D1 density provided the poorest growth conditions. This suggests that an appropriately managed bamboo density under a long-term bamboo-stocking retention strategy can enhance the shoot production of managed bamboo forests.

## 5. Conclusions

Stand density has a significant regulatory effect on the growth and soil quality of bamboo forests under a long-term bamboo-stocking retention model. Maintaining a stand density of 2400 ± 100 plants·hm^−2^ can effectively improve soil fertility, promote root development, and maximize aboveground yield. In contrast, a low density of 1200 ± 100 plants·hm^−2^ will lead to a significant decline in bamboo growth. In forestry practice, the optimal density standard can be directly used to optimize the sustainable management of bamboo forests. In the future, it will be necessary to carry out long-term monitoring in combination with annual age–class dynamics, to deeply analyze the formation mechanism of bamboo shoot yield, and verify the adaptability of the density model in different regions, so as to provide scientific support for the promotion of the results.

## Figures and Tables

**Figure 1 biology-14-01179-f001:**
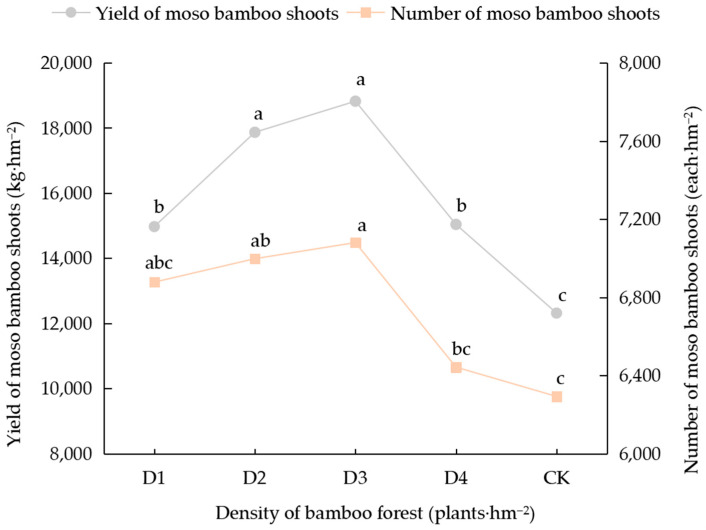
Bamboo shoot yield and the number of bamboo shoots under different moso bamboo forest stand densities. Note: Different lowercase letters in the figure indicate significant differences between different densities for the same data type (*p* < 0.05).

**Figure 2 biology-14-01179-f002:**
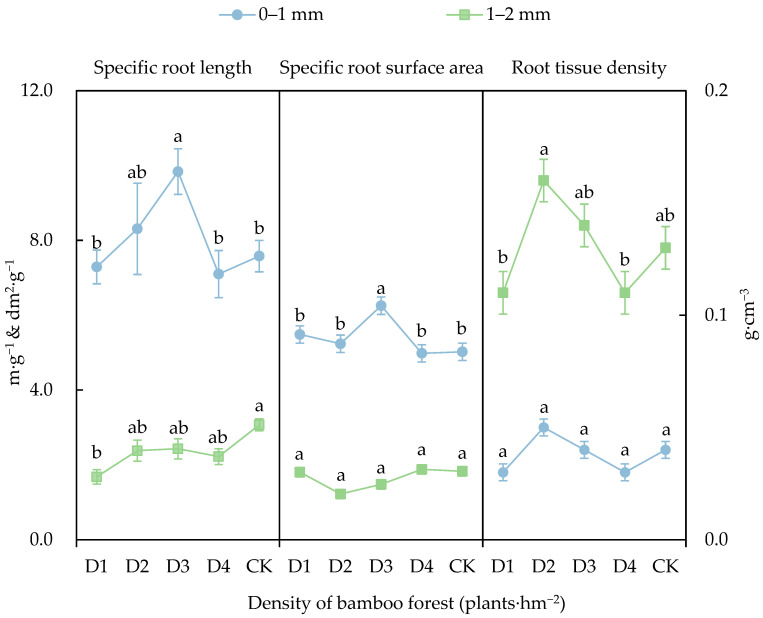
Morphological characteristics of moso bamboo roots under different density moso bamboo forest treatments. Note: Different lowercase letters in the figure indicate significant differences between different densities for the same data type (*p* < 0.05).

**Figure 3 biology-14-01179-f003:**
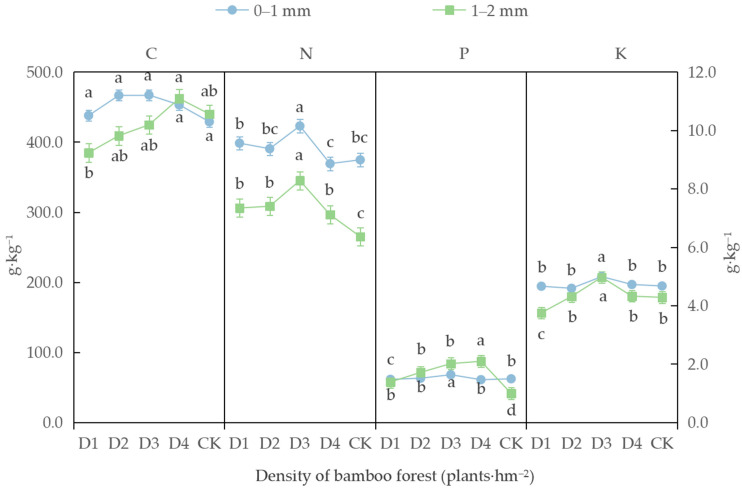
Nutrient content of moso bamboo roots under different moso bamboo forest density treatments. Note: Different lowercase letters in the figure indicate significant differences between different densities for the same data type (*p* < 0.05).

**Figure 4 biology-14-01179-f004:**
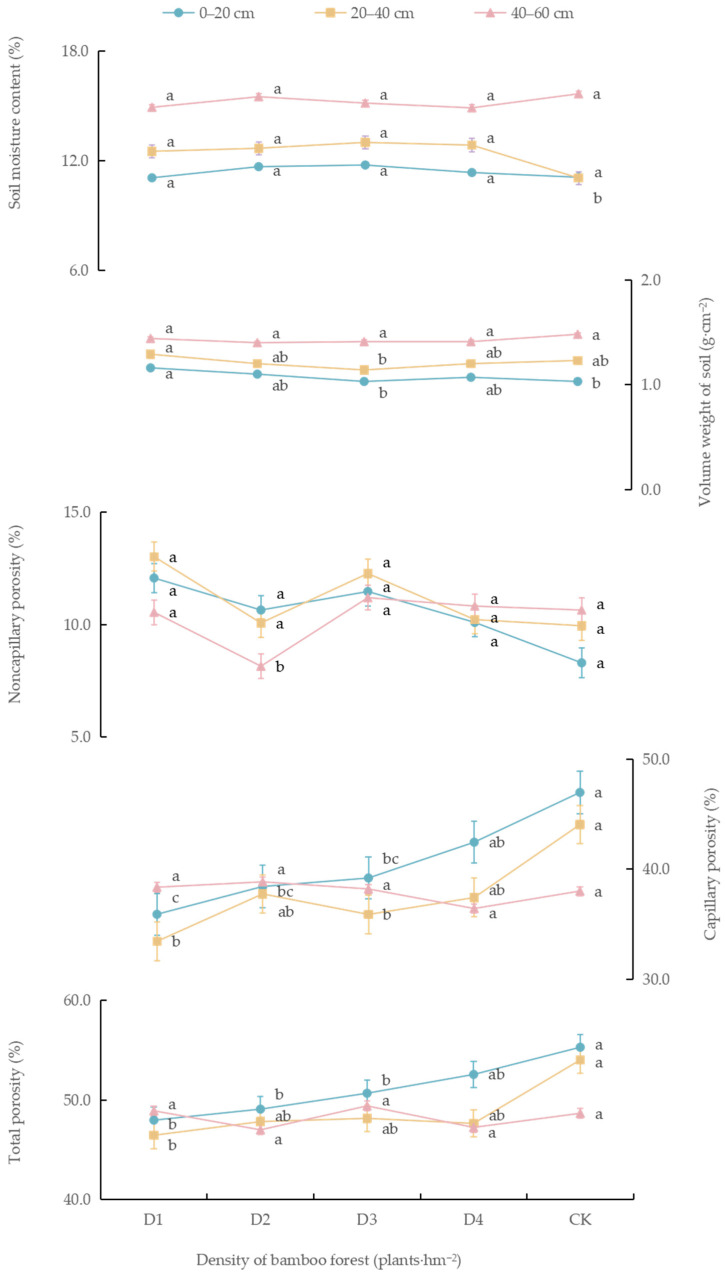
Soil physical properties of moso bamboo forest under different densities. Note: Different lowercase letters in the same column indicate significant differences in soil physical properties between different densities in the same soil layer (*p* < 0.05).

**Figure 5 biology-14-01179-f005:**
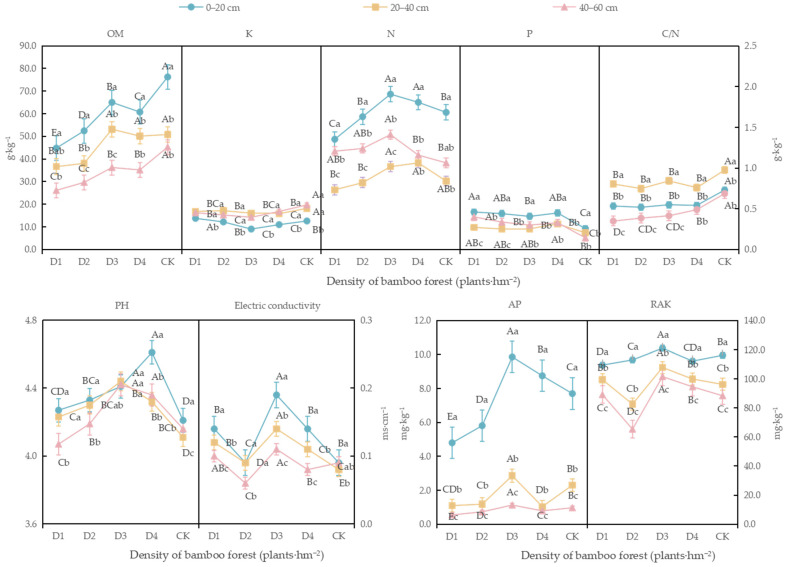
Soil chemical properties of moso bamboo forest under different densities. Note: Different capital letters within a figure panel figure indicate that there were significant differences between different densities in the same soil layer (*p* < 0.05). Different lowercase letters in the same figure panel indicate that the soil chemical properties of the same stand density significantly differed between soil layers (*p* < 0.05).

**Figure 6 biology-14-01179-f006:**
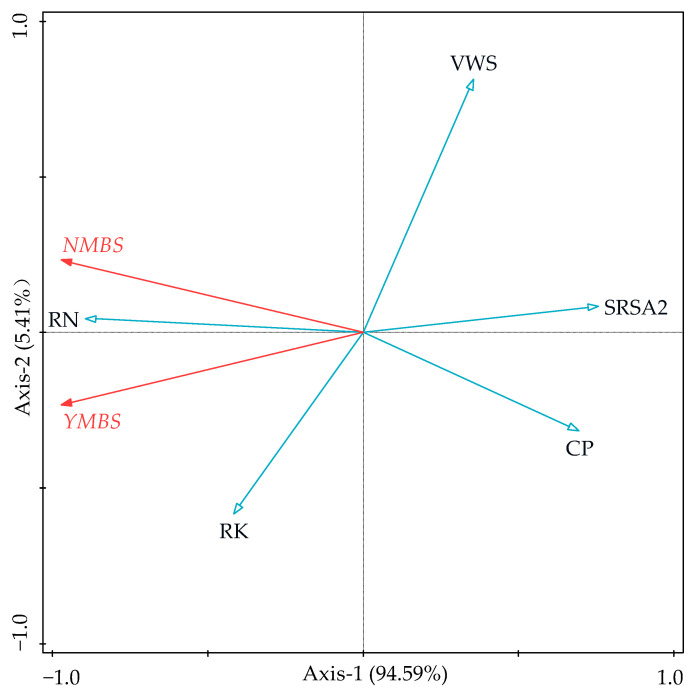
Redundancy analysis of root, soil quality, and shoot traits of moso bamboo. Note: YMBS, moso bamboo shoot yield; NMBS, number of moso bamboo shoots; SRSA2, specific root surface area 2; RN, root nitrogen content; RK, root potassium content; VWS, volume weight of soil; CP, capillary porosity.

**Figure 7 biology-14-01179-f007:**
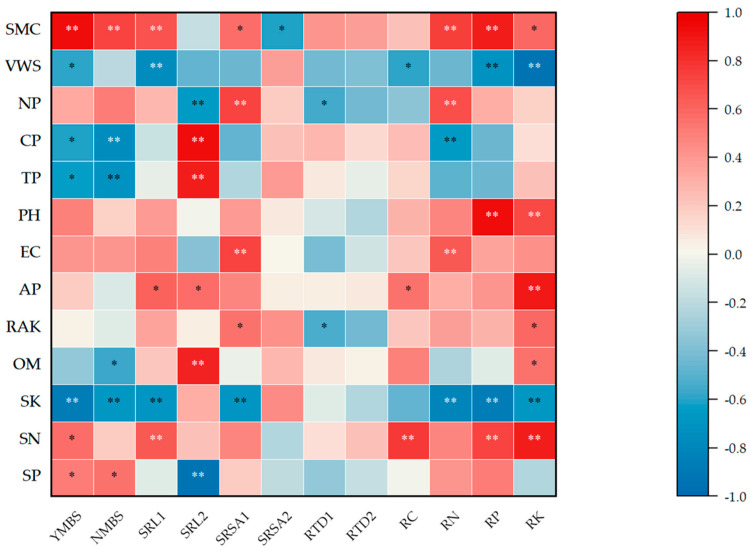
Heatmap of correlations between growth traits of moso bamboo and soil quality. Note: * represents a significant correlation (*p* < 0.05) ; ** indicated an extremely significant correlation (*p* < 0.01). YMBS, moso bamboo shoot yield; NMBS, number of moso bamboo shoots; SRL1, specific root length 1; SRL2, specific root length 2; SRSA1, specific root surface area 1; SRSA2, specific root surface area 2; RTD1, root tissue density 1; RTD2, root tissue density 2; RC, root carbon; RN, root nitrogen; RP, root phosphorous; RK, root potassium; SMC, soil moisture content; VWS, volume weight of soil; NP, non-capillary porosity; CP, capillary porosity; TP, total porosity; PH, pH value; EC, electroconductivity; AP, available phosphorous; RAK, readily available potassium; OM, organic matter content; SK, soil potassium; SN, soil nitrogen; SP, soil phosphorous.

**Table 1 biology-14-01179-t001:** Eigenvector, eigenvalue, contribution rate and cumulative contribution rate of the principal components.

Index	Principal Component					Eigenvector				
	1	2	3	4	5	1	2	3	4	5
Yield of moso bamboo shoots	0.93	−0.16	−0.34	−0.02	0.02	0.29	−0.06	−0.16	−0.01	0.02
Number of moso bamboo shoots	0.77	−0.45	−0.33	0.30	−0.02	0.24	−0.17	−0.16	0.20	−0.02
Specific root length 1	0.82	0.33	−0.22	0.41	−0.05	0.26	0.13	−0.10	0.27	−0.04
Specific root length 2	−0.24	0.92	−0.23	0.11	−0.18	−0.08	0.35	−0.11	0.08	−0.16
Specific root surface area 1	0.83	−0.01	0.17	0.52	−0.13	0.26	0.00	0.08	0.35	−0.11
Specific root surface area 2	−0.50	0.06	0.83	−0.11	−0.22	−0.16	0.02	0.39	−0.08	−0.20
Root tissue density 1	0.18	0.31	−0.91	0.17	−0.10	0.06	0.12	−0.43	0.11	−0.09
Root tissue density 2	0.28	0.22	−0.86	0.19	0.31	0.09	0.08	−0.41	0.12	0.28
Carbon content in roots	0.28	0.52	−0.04	−0.55	0.59	0.09	0.20	−0.02	−0.37	0.53
Nitrogen content in roots	0.92	−0.18	0.03	0.33	−0.11	0.29	−0.07	0.01	0.22	−0.10
Phosphorus content in roots	0.84	0.04	0.14	−0.48	−0.22	0.26	0.02	0.07	−0.32	−0.20
Potassium content in roots	0.74	0.63	0.12	0.07	−0.19	0.23	0.24	0.06	0.05	−0.17
Soil moisture content	0.87	−0.06	−0.32	−0.27	−0.25	0.27	−0.02	−0.15	−0.18	−0.22
Volume weight of soil	0.66	0.70	−0.09	−0.23	−0.12	0.21	0.27	−0.04	−0.15	−0.10
Non-capillary porosity	0.50	−0.48	0.61	0.38	−0.03	0.16	−0.18	0.29	0.25	−0.02
Capillary porosity	−0.56	0.82	−0.13	0.02	−0.03	−0.17	0.31	−0.06	0.01	−0.03
Total porosity	−0.47	0.83	0.16	0.24	−0.02	−0.15	0.32	0.08	0.16	−0.01
PH value	−0.67	−0.22	−0.28	0.48	0.44	−0.21	−0.09	−0.13	0.32	0.40
Electric conductivity	0.63	−0.11	0.57	0.28	0.43	0.20	−0.04	0.27	0.19	0.39
Available phosphorus	0.47	0.82	0.31	0.07	0.06	0.15	0.31	0.15	0.05	0.05
Rapidly available potassium	0.38	0.36	0.81	0.25	0.15	0.12	0.14	0.38	0.17	0.13
Organic matter content	−0.10	0.96	0.23	0.07	0.09	−0.03	0.37	0.11	0.05	0.08
Soil potassium content	−0.97	0.00	−0.07	0.18	−0.18	−0.30	0.00	−0.03	0.12	−0.16
Soil nitrogen content	0.75	0.56	0.13	−0.25	0.22	0.23	0.21	0.06	−0.17	0.20
Soil phosphorus content	0.45	−0.78	0.09	−0.39	0.15	0.14	−0.30	0.04	−0.26	0.13
Eigenvalue	10.289	6.846	4.406	2.215	1.244	10.289	6.846	4.406	2.215	1.244
Contribution %	41.16	27.38	17.62	8.86	4.97	41.16	27.38	17.62	8.86	4.97
Cumulative contribution %	41.16	68.54	86.16	95.02	100.00	41.16	68.54	86.16	95.02	100.00

**Table 2 biology-14-01179-t002:** Comprehensive evaluation of the growth potential of moso bamboo forests under different stand densities.

Treatment/(Plant·hm^−2^)	Principal Component Score					Comprehensive Score	Sort
	F1	F2	F3	F4	F5		
D1	−3.59	7.54	−0.92	0.90	−0.53	0.48	5
D2	−3.44	9.31	−2.85	0.45	−0.70	0.64	3
D3	−1.93	9.89	−1.29	1.14	−0.58	1.76	1
D4	−3.89	9.84	−0.99	−0.14	−0.56	0.88	2
CK	−5.89	12.14	−1.79	0.97	−0.74	0.63	4

## Data Availability

The original contributions presented in this study are included in the article. Further inquiries can be directed to the corresponding author(s).

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
