# Peer review of "Effects of Moso Bamboo (Phyllostachys edulis) Forest Stand Density on Root Growth and Soil Quality for Shoot Production Under a Long-Term Bamboo-Stocking Retention Model"

_biology, 2025, doi:10.3390/biology14091179_

Round 1

Reviewer 1 Report

Comments and Suggestions for Authors

Review on „Effect of moso bamboo (Phyllostachys edulis) forest stand density on root growth and soil quality for shoot production under a long-term stocking bamboo retention model” by He et al.

This manuscript presents a well-structured, comprehensive study investigating how different stand densities in a long-term stocking bamboo retention management model affect moso bamboo shoot yield, root traits, and soil quality. The research is timely and relevant, especially considering the growing demand for sustainable forest and bamboo shoot production practices. The findings are supported by data analysis, including correlation, redundancy, and principal component analyses. The manuscript is suitable for publication after minor revisions aimed at improving clarity, consistency, and presentation.

The study introduces a new management paradigm focusing on 1- to 3-year-old stocking bamboo while suppressing new shoots, which differs from traditional thinning models. It provides an evidence-based recommendation (2400±100 plants·hm⁻²) for optimal stand density, with substantial implications for sustainable bamboo cultivation and land management. The experimental design includes well-defined treatments and a traditional control. Multiple biological and physicochemical parameters were measured across soil depths and bamboo tissues, offering a multidimensional understanding of the system. The manuscript provides actionable guidelines for forest managers and farmers regarding ideal planting density for maximizing bamboo shoot yield without compromising soil health.

Specific comments:

The manuscript is generally readable but would benefit from professional language editing to address awkward phrasings, grammar inconsistencies, and occasional redundancies. Example:

...was at its maximum (9.84 cm g⁻¹) please consider: ...reached its maximum value (9.84 cm g⁻¹)

Title:

Please consider clarifying “Long-Term Stocking Bamboo Retention Model” for a global audience unfamiliar with the term.

Abstract:

The abstract is informative but could be rephrased for conciseness and clarity. e.g....both bamboo shoot yield and the number of shoots first increased and then decreased with forest density. could be:  Bamboo shoot yield and number exhibited a unimodal response to stand density.

Line 18: The authors can’t state “the best way” since they examined the bamboo density so no information about other circumstances. Please correct it.

The discussion is well-researched but slightly repetitive, particularly in reiterating known relationships (e.g., nitrogen's role in plant metabolism), please double-check and rewrite where necessary.

Some citations are outdated or not clearly integrated into the discussion. Where possible, refer to more recent and international studies on bamboo or related perennial crops.

Line 562-568. The WinRHIZO software is capable of measuring and evaluating more parameters than those presented in the manuscript. Please explain why certain data, such as root length by root diameter class, were not included. I believe this information could provide additional insights and strengthen the study.

Use consistent formatting units throughout the manuscript (e.g., “plants·hm⁻²” vs. “plants/ha”).

Comments on the Quality of English Language

The manuscript is generally readable but would benefit from professional language editing to address awkward phrasings, grammar inconsistencies, and occasional redundancies. 

Author Response

1.Introduction explains what is ' Long-Term Stocking Bamboo Retention Model '.

2.Abstract : We modified it according to your opinion.

3.We modified the problem raised in Line 18.

4.Discussion : In view of the previous problems, such as the short discussion, the lack of in-depth discussion of the ecological or practical effects of the research results, and the lack of discussion of the correlation between root nitrogen content and aboveground yield, we modified it.

5.2.3.Test methods : This study focused on the correlation between root architecture and density response. Therefore, among the parameters generated by WinRHIZO, we preferentially selected parameters directly related to physiological functions ( such as specific root length, specific surface area and root tissue density ).

6.Consistent format units have been used in the full text ( e.g., 'plants·hm⁻²' and 'plants/ha' ).

Reviewer 2 Report

Comments and Suggestions for Authors

Brief Summary

Dear authors:
In accordance with COPE's ethical guidelines for academic reviewers, I am pleased to submit this review report on the aforementioned manuscript. I have carefully read the document and would like to thank the authors for their efforts. The study addresses an important issue regarding the optimization of planting density in Phyllostachys edulis bamboo forests to maximize shoot production, improve root growth, and maintain or improve soil quality within a long-term standing bamboo retention model. The experiment was conducted in Fujian Province, China, comparing four different planting densities (1200, 1800, 2400, and 3000 plants·ha⁻¹), in addition to a traditional control (2100 plants·ha⁻¹). Variables related to shoot growth, root morphology, root nutritional content, and physical and chemical properties of the soil were analyzed. The results indicate that a density of 2400 plants·ha⁻¹ is optimal for maximizing shoot production (18.822 kg·ha⁻¹), with significant improvements in soil quality and root morphological and nutritional characteristics. Nitrogen content in roots explained 75.7% of the variation in yield, demonstrating its decisive role.

The manuscript represents a valuable contribution to sustainable forest management, particularly in bamboo production systems, and offers relevant practical implications for improving productivity without compromising soil health.

Specific comments
In my opinion, the manuscript has an atypical structure, placing the results section before the methodology. This inversion contravenes the conventional IMRyD (Introduction, Methodology, Results, and Discussion) format, which may hinder the reader's understanding and should be corrected to improve the flow and logic of the scientific text.
1. Title
The title is appropriate and contains the essential epistemological elements: object of study (root growth, soil quality, shoot production), subject of study (Phyllostachys edulis), and method (Effects). However, it could be shortened for greater clarity, for example:
1.- Effects of Stand Density for Shoot Production and Soil Quality in Moso Bamboo (Phyllostachys edulis) Forests. 15 words, however, the suggestion mentioned is up to the authors.

2. Abstract
Remember that this section is the most widely read part of a manuscript. The abstract adequately fulfills the basic functions of this section: it provides an overview of the problem, the methods used, the most relevant results, and a conclusion. However, the text leans toward a narrative of results without clearly separating the fundamental parts (introduction of the problem, objective, methodology, results, and conclusion), which would make it difficult for those who only consult the abstract to read it quickly and in a structured manner. Furthermore, although the effects of density on roots, soil, and shoots are mentioned, the objective is not explicitly presented as a separate sentence. 

It is also relevant to note that the use of the term “discovered the best way” suggests an absolute statement that is inappropriate for scientific writing. It would be preferable to maintain a more cautious, evidence-based wording (“suggests that...”). On the other hand, the language is clear and accessible to specialized readers, and technical terms are used correctly. 
The abstract could be improved by explicitly organizing its sections (a brief introduction to the field of knowledge, problem, objective, methodology, results, conclusion) into separate paragraphs or sentences, maintaining an objective and formal style.

Simple Summary:

In my opinion, this section is ambiguous in terms of social impact or value to society. It mentions that the study “provides a scientific basis,” but does not specify how this directly benefits society (e.g., food security, soil conservation, producer income). It is suggested that the practical value be explicitly mentioned: “This information could help bamboo farmers improve yield while protecting soil quality.”  The sentence “This study provides a scientific basis...” is ambiguous for a general reader. It can be rephrased with direct impact: “This research helps guide better planting strategies for farmers.”

The Simple Summary partially meets the criteria established by the journal. It is concise, clear in its intent, and without excessive technical jargon, but it needs improvement in the translation of technical language and in the explanation of the social benefit.

The Simple Summary partially meets the criteria established by the journal. It is concise, clear in its intent, and without excessive technicalities, but it needs improvement in the translation of its technical language and in the explanation of the social or practical benefit.

3. Keywords
It is recommended to avoid repeating terms from the title and abstract. Concepts related to sustainability could be included, for example:
Sustainable forest management, nutrient cycling, soil plant interactions, Related SDG.

4. Introduction
The introduction to the manuscript establishes a relevant general context regarding the ecological and economic importance of Phyllostachys edulis bamboo in Asian forest systems. It also presents a clear justification for the need to explore long-term retention models as opposed to traditional intensive management. However, critical analysis reveals that this section lacks a structured transition between the general problem (overexploitation of soil resources) and the specific problem (optimization of densities in the LR model). 

The knowledge gap is not explicitly stated, nor is a hypothesis or research question formulated to guide the reader toward the objectives of the study. Likewise, the literature review is relevant but superficial; I invite the authors to conduct a brief review of the evidence (what has been done, what has not been done, and what will be done in the manuscript). A more extensive discussion of previous studies analyzing the effect of planting density on root growth and soil quality would be desirable. This would allow the original contribution of the study to be better highlighted. 

Finally, the introduction lacks a clear statement of the objective at the end of the section, which is essential to guide the reader on the specific purpose of the research. In summary, although the introduction provides good contextualization, it would benefit from a more argumentative structure and greater critical depth.

It can be strengthened with the following adjustments:
1.-Present the problem and its justification for the study in a clear and separate paragraph.
2.-A literature review section (empirical evidence) 
3.-Explicitly identify knowledge gaps.
4.-Write the objective explicitly at the end of the section, including the working hypothesis if possible.

5.-Objectives and hypotheses
The general objective is implicit, but it must be clearly stated in all sections, in the abstract, and in the introduction, and it must be aligned with the title of the paper.
The manuscript does not contain a specific section or a direct statement of the general objective of the study. This represents a significant methodological omission, as all experimental designs must be aligned with clearly defined objectives. Likewise, the hypothesis is not formulated at any point in the text, either explicitly or implicitly, which weakens the study's argumentation and statistical justification. 

A hypothesis such as “An intermediate planting density is expected to maximize shoot production without compromising soil quality” would have been useful in guiding the interpretation of results. 

In applied research such as this, where agronomic, forestry, and soil variables intersect, it is crucial to precisely define the what and why of the measurement. This absence affects the conceptual clarity of the article, hinders the reproducibility of the experimental reasoning, and limits the discussion of the findings. 

For future work, it is recommended to include a paragraph at the end of the introduction that clearly articulates the research question, the general objective, and the hypothesis(es) to be tested.

It would be desirable to add a working hypothesis, since it presents empirical work and also performs a statistical analysis where statistical hypotheses are proposed.

6. Methodology
The methodology section is well organized and details the site conditions, experimental design, and variables analyzed. Five treatments with different planting densities were used, which is appropriate for gradually comparing the effects on the parameters studied. However, there are several opportunities for improvement: 

1.-There is no diagram or table relating variables, units of measurement, sampling frequency, and analytical methods, which would be useful for replicability. 
2.-The description of the statistical methods is described in the data analysis section; although the use of ANOVA and post hoc tests (LSD), principal component analysis, heat analysis, and Pearson correlation are mentioned, the criteria for normality, homogeneity of variances, or transformations applied are not indicated, which are fundamental aspects for validating the comparisons made. 
4.-Furthermore, the instruments used for nutrient, pH, or moisture measurements are not clearly mentioned, which could affect the transparency of the procedure. 
Although the experimental design seems adequate, its presentation is insufficient in technical terms. 

Finally, there is no ethical reflection on the management of the forest system or indications as to whether the study was subject to any research authorization. 
In my opinion, the methodology needs to be strengthened in terms of technical transparency and statistical rigor.

7. Results and Discussion 
The results section is presented clearly, with well-designed figures and tables that are relevant to the study objectives. The trends found, such as improved shoot production and soil quality at intermediate densities, are consistent with expectations and are interpreted appropriately in the text. However, the discussion is brief and does not delve into the ecological or practical implications of the findings. 

Critical analysis reveals a limited discussion of the contrast with previous literature, omitting to analyze why the effects found were observed or how they relate to the physiological mechanisms of bamboo. Likewise, the limitations of the study are not made explicit, which is essential in fieldwork with inherent variability.  And many (sections).

The correlation between nitrogen content in roots and shoot yield is a noteworthy finding that deserves further analysis. 
Implications for forest management are also not discussed, which represents a missed opportunity. 

I suggest merging the results and discussion sections to allow for critical scientific reflection. It would be desirable to structure this section according to a classic outline: comparison with previous studies, interpretation of mechanisms, practical implications, and limitations. 
In their current form, the results are useful but underutilized from an academic standpoint.

In section 2.2.1. Morphological characteristics of bamboo roots
It should be clear that the morphological characteristics analyzed are part of an evidence-based classification. As presented in the manuscript, it could be considered incomplete or poorly justified, especially for the following reasons:

The title suggests a comprehensive analysis of the morphological characteristics of the roots, but in reality only three variables are described. These three characteristics, although relevant, do not exhaust the concept of root morphology, which usually also includes:
Density of fine vs. coarse roots,Number of lateral roots,Anchoring depth,Distribution by soil layers,Root/dry weight ratio or volume of soil explored. Therefore, the content does not fully justify the title.

There is no explanation as to why these three characteristics were specifically chosen and not others. A more reasoned selection (physiological, agronomic, or ecological) would strengthen the section.

These morphological characteristics are not clearly linked to relevant functions of bamboo, such as nutrient absorption, competition for water, or resilience to management. Without this connection, the section remains more of a description than an analysis. 

If the objective is to recommend an optimal density for sustainable production, how do these three variables help in making practical decisions? This is not entirely clear in the section.
Therefore, I suggest changing the title to something more precise, such as:

 Key root morphological traits under different stand densities

Or Expand the analysis if the data allow, including other relevant morphological variables.
Better justify the choice of variables and discuss how they relate to yield, nutrient uptake, or soil quality. 
The authors should explain why they chose these characteristics in the study limitations section.

Section 2.2.2: Nutrient content of bamboo roots

The study analyzed the content of: Nitrogen (N), Phosphorus (P), Potassium (K), Carbon (C).
Of these, N, P, and K are macronutrients, essential for plant growth, while carbon is more of an indicator of the organic/structural content of the roots.

Is this sufficient? From a strictly agronomic perspective, including N, P, and K is reasonable, as they are essential for tissue growth, photosynthesis, and cellular metabolism. In many plant species, deficiencies in these elements are correlated with reduced biomass and yield. However, in a study that aims to optimize soil quality and long-term root development in bamboo, a more comprehensive view would be expected; limiting oneself to NPK could oversimplify the nutritional diagnosis.

Remember Liebig's Law of the Minimum, which states that plant growth is limited by the essential nutrient that is in the lowest relative availability, not necessarily by N, P, or K. In the case of bamboo, studies have documented that calcium (Ca), magnesium (Mg), sulfur (S), iron (Fe), zinc (Zn), or other micronutrients may also be limiting, depending on soil type, climate, and management system. Therefore, restricting the analysis to NPK alone does not allow for the proper application of the law of the minimum, as it is not possible to identify which nutrient is truly limiting if others were not measured.

The authors should explain why they chose these elements in the limitations section of the study. 

Carbon as an additional indicator is useful as a marker of root organic content, but it is not a limiting nutrient per se. Its inclusion here is more structural than functional.
I suggest the following to the authors:
The choice of N, P, K, and C is adequate as a baseline, but it is insufficient if the purpose is to identify nutritional limitations or apply the law of the minimum.
For a more robust study applicable to sustainable bamboo management, it is recommended to include at least the secondary macronutrients (Ca, Mg, S) and one or two key micronutrients (Fe, Zn, or boron).
It would be very useful to justify why other nutrients were excluded, or to acknowledge this as a methodological limitation in the discussion, or to suggest future lines of research in this regard.

Section 2.3.2 Bamboo forest soil chemical properties

In this section, the study analyzes five chemical properties of the soil: The variables chosen are common and suitable for describing general soil fertility.
pH: level of acidity or alkalinity, which regulates nutrient availability.
OM: organic matter content, key to water and nutrient retention.
Three essential nutrients (N, P, K), necessary for bamboo growth.

This set of variables allows for a direct comparison with the nutritional content of the roots, which is useful for exploring relationships between soil and plant.
These variables are inexpensive and easy to measure in the field and laboratory, which facilitates their use in practical monitoring.

Limitations and critical omissions:
Absence of important basic cations, explain why Calcium (Ca), magnesium (Mg), and cation exchange capacity (CEC) were not evaluated, which are essential for:
 pH stability, enzyme activity, and root development.

Because electrical conductivity (EC) and trace elements (such as aluminum or exchangeable sodium), which can directly affect bamboo growth, especially in high-density systems, were not included.

Why was the C/N ratio not evaluated? Although carbon and nitrogen were measured separately, the C/N ratio was not reported, despite being a critical indicator of: nitrogen availability, organic matter decomposition dynamics, and soil microbial activity.

 In soil health studies, especially in forest systems, it is recommended to include indicators such as dehydrogenase or phosphatase activity and microbial biomass. Their omission limits the understanding of key biogeochemical processes under different bamboo densities.

 Despite these limitations, the selected variables are adequate as a baseline for characterizing soil chemical fertility, but the analysis is too limited to comprehensively assess soil quality in a long-term retention model. In particular, the exclusion of cations, C/N ratio, and microbial activity weakens the ecological diagnosis of the system and its applicability in sustainable management programs.

The selection of variables in section 2.3.2 allows for a basic characterization of soil fertility, but is limited in comprehensively assessing soil quality under a long-term forest management model. 
Therefore, changing the title would be appropriate.

For future lines of research, it would be advisable to include additional variables such as calcium, magnesium, C/N ratio, and at least one indicator of microbial or enzymatic activity, in order to better capture the processes of regeneration and soil sustainability in high-density Phyllostachys edulis systems.

Section 2.6 PCA:
This section presents a principal component analysis (PCA) to identify the most relevant factors affecting the production of Phyllostachys edulis shoots under different planting densities. In general terms, the methodological choice of PCA is relevant, as it allows for the reduction of data dimensionality and the detection of patterns of covariation between multiple physiological and edaphic variables. However, there are several points that should be critically discussed in relation to the selection of variables, the interpretation of results, and their practical applicability.

The use of principal component analysis is appropriate for synthesizing multivariate information and discovering latent correlations between variables that affect shoot production. PCA integrates data on roots, soil, and nutritional content, which gives the study coherence and allows for a multicausal evaluation of the effects of planting density.

The finding that root density, root nitrogen content, soil pH, and organic matter are strongly correlated with shoot production is consistent with the literature on bamboo growth.However, I would like to point out some issues that I suggest could be addressed:  As discussed above, the study only includes a small subset of variables:
Morphological variables of roots,  soil, and  roots (nutrients): N, P, K, C

This limits the explanatory power of PCA, as it excludes key edaphic variables such as calcium, magnesium, cation exchange capacity (CEC), C/N ratio, microbial activity, and salinity. These variables could substantially modify the interpretation of PCA if they were incorporated.

 The manuscript does not discuss in depth what each principal component represents from a biological or agronomic point of view. 

What process or function do PC1 and PC2 summarize? 

Is it soil fertility, root efficiency, or soil-plant interaction?

In addition, there is a lack of statistical or technical justification, as the criteria for selecting variables for PCA are not mentioned (were all variables included? Was there a prior reduction?). 
Nor is it indicated whether the data were standardized or whether the adequacy of the matrix was verified (e.g., Kaiser-Meyer-Olkin tests or Bartlett's sphericity).

Finally, although PCA clearly identifies variables that explain yield, the manuscript does not translate these findings into practical management recommendations. This limits its applicability for forestry technicians or producers.

The inclusion of principal component analysis is a methodological strength of the study. However, its interpretive and practical value is restricted by an incomplete selection of variables and a superficial functional discussion. 

PCA reinforces previous findings on the importance of root density and nitrogen, but leaves out key dimensions of soil quality and the rhizosphere ecosystem. To improve this section, it would be essential to:  Expand the set of variables included.
 Better explain the ecological significance of the principal components.
Translate the results into concrete recommendations for density management in bamboo plantations.

9. Conclusions
The conclusions section correctly summarizes the most important findings of the study, highlighting that a density of 2400 plants·ha⁻¹ represents the optimal condition for maximizing shoot production and improving soil quality in Phyllostachys edulis plantations under a prolonged retention model. However, the conclusions are presented more as a descriptive summary than as an integrative reflection. It does not make explicit how the findings relate to the objectives set (which, as mentioned, were not clearly formulated either). 

Furthermore, the section does not clearly point out the practical, theoretical, or policy implications of the results, nor does it mention the limitations of the study, aspects that are essential in applied studies with potential impact on forestry or agricultural management policies. 
Nor does it offer a clear proposal for future research, which detracts from the continuity of the line of study. 

From a scientific perspective, a solid conclusion should critically synthesize the findings, discuss their applicability, and point out the margins of uncertainty. 
In its current form, the section serves to close the document, but it does not provide the added value that a well-developed conclusion can offer.
The conclusions are consistent with the results. I suggest:
1.-Briefly reiterate the objective and how it was achieved.
2.-Explicitly point out practical applications and future lines of research.
3.-Practical, theoretical, or policy implications.

10.References
The list of references covers an adequate bibliographic base, with relevant articles on bamboo physiology, soil dynamics, and forest management. However, critical analysis reveals certain aspects that could be improved. 
1.-First, the proportion of recent references (last 5 years) is moderate, but a greater presence of updated international literature would be expected, particularly on sustainable bamboo management models. 
2.-Second, most of the sources are Chinese studies (probably due to geographical affinity), which is understandable but may limit the comparative perspective with other forestry systems. 
3.-The review does not incorporate international policy sources, sustainability criteria (e.g., FAO, IPBES), or broader conceptual frameworks on adaptive management, which would have provided an interdisciplinary perspective. 
4.-There is no inclusion of literature on advanced statistical methodologies or justification of soil quality indicators, which would have strengthened the technical basis of the study. 
5.-Finally, the list of references is well organized and follows the journal's format, but the article would gain in depth if more sources were added to support the discussion and the approach to the problem.

Author Response

1.Abstract, Simple Summary and Keywords section, we modified according to your comments. 2.Introduction: we have modified the introduction of bamboo, added hypotheses, and proposed scientific problems to be solved. 3.Methodology: We modified it according to your opinion and added the main instruments used. We point out the criteria of normality, variance homogeneity or application transformation, and describe variables, measurement units, sampling frequency and analysis methods in detail. 4.Results: We modified it according to your opinion. We added the data of C / N ratio and modified the title. Thank you very much for your profound opinions on the limitations of soil analysis. Due to the end of the research cycle and the lack of relevant indicators in the original scheme, we cannot backtrack and supplement the data of Ca, Mg, CEC, EC, trace elements and microorganisms, and we apologize for this. 5.Discussion In view of the previous problems, such as the short discussion, the lack of in-depth discussion of the ecological or practical effects of the research results, and the lack of discussion of the correlation between root nitrogen content and aboveground yield, we modified it. 6.Conclusions We have made the following modifications : (1)Briefly reaffirm the goal and its implementation. (2)The practical application and future research directions are clearly pointed out. (3)Practical impact.

Please see the attachment for a more detailed response.

Reviewer 3 Report

Comments and Suggestions for Authors

I thank the authors for presenting this useful study.

Please respond to the following comments and notes, which aim to improve the manuscript and provide a full understanding of all aspects of the work without detracting from its scholarly value.

1- The abstract needs some editing and shortening, as the characteristics and results of the 2,400 plant/h treatment, were over-explained, considering it the best treatment, while the results of other treatments, especially the lowest, were ignored. Also, add at the end of the abstract, a recommendation summarizing the final results.

2- There seems to be a mistake on line 35, there is no need to write m2 since the unit is plant/ha.

3- The authors mentioned a "long-term" regimen several times in the abstract and introduction (line 107) , but they did not mention the duration or what is meant by long-term. Therefore, I did not see sufficient justification for conducting this experiment, nor did I see any significant addition compared to previous reference studies, many of which the authors cited, that summarized the same results as the current study.

4- In figure 1, there is no need for the decimal point since the numbers are large on one hand and the value zero after the decimal point has no meaning.

5- In Material and Methods, line 549, three plants were selected. Isn't relying on measurements from "three plants" too small for the size of the treatment? What is meant by "three representative plants"? What is the standard?
6- On what basis was the radius of the circle (60 cm) from which the soil samples were collected determined?

Thanks

Author Response

1.Abstract : We modified it according to your opinion and added a suggestion to summarize the final results at the end of the summary.

  1. Specify the specific meaning of duration or ' long-term ' in the Introduction.

3.In Figure 1, the decimal point is removed.

4.In materials and methods, our sampling number and methods are designed with reference to previous studies, and relevant references have been added.

Round 2

Reviewer 2 Report

Comments and Suggestions for Authors

Dear authors, 
The proposed changes and suggestions were addressed as far as possible given the limitations of the study. I strongly recommend that this section be included separately. The guidelines for responding to reviewers must be discussed on scientific grounds. In science, we are all open to debate and must respond appropriately, preferably with references.  
The manuscript requires improvements in the editing of the graphics. The editorial office requests a resolution of 400 dpi for better visualization.
Therefore, after analyzing the authors' changes and responses, I recommend accepting the manuscript with minor revisions.